# A Multiplex PCR Assay for Differential Identification of Wild-type and Vaccine Strains of *Mycoplasma gallisepticum*

**DOI:** 10.3390/pathogens12010111

**Published:** 2023-01-09

**Authors:** Sung-Il Kang, O-Mi Lee, Hye-Jin Lee, Yong-Kuk Kwon, Myeong Ju Chae, Ji-Yeon Jeong, Min-Su Kang

**Affiliations:** Avian Disease Research Division, Animal and Plant Quarantine Agency, Gimcheon-si 39660, Republic of Korea

**Keywords:** *Mycoplasma gallisepticum*, chicken, live vaccine, multiplex PCR, diagnosis, differentiation

## Abstract

*Mycoplasma gallisepticum* (MG) can cause respiratory disease in chickens and result in serious economic losses in the chicken industry. The use of live vaccines has been a favorable option for the control of MG infection in multi-age commercial layers and broiler breeders. There are three live vaccines, including ts-11, 6/85, and F strain, that have been commonly used in various parts of the world, including South Korea. The definitive diagnosis of the infection, therefore, requires the differentiation of wild-type field strains of MG from the vaccine strains used. Thus, we aimed to develop a novel multiplex PCR assay to discriminate between vaccine strains (ts-11, 6/85, and F strain) and wild-type field strains of MG isolated from infected chickens. We designed four novel primer sets that are each specific to MG species, ts-11, 6/85, and F strain. The multiplex PCR assay using the primer sets differentially identified wild-type and vaccine strains of MG but did not detect other avian bacteria. The detection limit of this assay was 250 fg/μL of genomic DNA of each strain tested. In addition, this assay was applied to 36 MG strains isolated from chickens over the past 20 years in South Korea. As a result, the assay identified 22 wild-type strains and 14 vaccine strains. Consequently, the novel multiplex PCR assay can discriminate between vaccine and wild-type field strains of MG and could be a valuable tool for the diagnosis of MG infection in MG-vaccinated chicken flocks.

## 1. Introduction 

*Mycoplasma gallisepticum* (MG) is an avian pathogen that phylogenetically belongs to the class Mollicutes. This pathogen is known to cause chronic respiratory disease involving airsacculitis in chickens. Symptoms of MG infection include nasal discharge, sneezing, coughing, and conjunctivitis [1,2,3]. MG infection is also transmitted vertically through infected eggs and horizontally by direct or indirect contact with infected birds. It is also a worldwide problem, as it reduces egg production, hatchability, and feed efficiency on poultry farms, causing huge economic losses in the poultry industry in various parts of the world [4]. In addition, this infection may be exacerbated by complications with *E. coli* and other viral pathogens, such as infectious bronchitis virus, rather than a single infection [1]. Utilizing live vaccines has been a favorable option for the control of MG infection in multi-age commercial layers and broiler breeders in many countries, including South Korea [2,5,6,7]. Three live vaccines, including ts-11, 6/85, and F strain, are commercially available for preventing MG infection worldwide.

Diagnostic methods for MG infection include serological tests, such as serum plate agglutination test (SPA) or enzyme-linked immunosorbent assay (ELISA), bacterial cultures, and genetic methods using polymerase chain reaction (PCR) and real-time PCR [3,8,9,10,11]. However, the definitive diagnosis of the infection requires the differential identification of wild-type field strains of MG and vaccine strains in those areas where live MG vaccines are used. Currently, MG strains can be distinguished by the genotyping offered by gene-targeted sequencing (GTS) using four primers and multilocus sequence typing (MLST) using six housekeeping genes, allowing for the differentiation between wild-type and vaccine strains of MG [12,13,14]. However, this approach takes a very long time because MG strains are differentiated through PCR and subsequent sequence analysis.

Therefore, the purpose of this study was to develop a fast and accurate multiplex PCR assay that can differentiate wild-type MG strains from the three types of MG vaccine strains (ts-11, 6/85, and F strain) in a single reaction.

## 2. Materials and Methods

### 2.1. Bacterial Strains and DNA Extraction

This study included five reference strains (including three vaccine strains) and thirty-six field strains of MG, including twenty-two wild-type strains and fourteen vaccine strains (one ts-11, one 6/85, and twelve F strains), along with twenty-seven strains of other bacterial species (Table 1). The three vaccine strains, ts-11 (Vaxsafe^®^ MG, Boehringer Ingelheim., Seoul, South Korea), 6/85 (Nobilis^®^ MG 6/85, MSD Animal Health, USA), and F strain (PoulShot^®^ MG-F, CAVAC Inc., Daejeon, South Korea), were purchased from vaccine companies. MG field strains were isolated from the tracheas (tissues and cotton swabs) from chickens from different commercial poultry flocks during the period of 2005–2021. These isolates were confirmed by using PCR as reported in the WOAH standard diagnostic manual [3] and were also differentiated into wild-type strains and vaccine strains using GTS analysis [12].

DNA extraction from cultured strains was performed according to the manufacturer’s instruction using a DNA blood and tissue kit (Qiagen Korea Ltd., Seoul, South Korea), and the DNA was stored at −20 °C.

### 2.2. Sequence Analysis and Primer Design

To identify genetic regions specific to MG species and the three vaccine strains (ts-11, 6/85, and F strain), the whole-genome sequences of eighteen MG strains (GenBank accession nos. NZ_CP044225, NZ_CP044224, NC_017503, NZ_CP028146, NZ_ CP028147, NZ_CP044226, NC_023030, NC_004829, NC_017502, NC_018406, NC_018407, NC_018408, NC_018409, NC_018410, NC_018411, NC_018412, NC_018413, and NZ_LS991952) were downloaded from the NCBI genome database and analyzed using CLC GenomeWorkbench (ver. 20) software (Qiagen, Aarhus, Denmark). Candidate regions specific to each of the three MG vaccine strains and MG species were validated with a homology search using BLAST on the NCBI website. Four primer sets that are each specific to MG species, specifically strain ATCC 15302, as well as ts-11, 6/85, and F strain, were designed using CLC Main Workbench 20. These primer sequences and the sizes of the amplification products are listed in Table 2.

### 2.3. Multiplex PCR Assay

A multiplex PCR assay was performed to discriminate between the three MG vaccine strains and the field strains using four target regions (*parE*, RS03710-rlmB, *scpA*, and MGF_RS03965). PCR amplification was optimized using AccuPower^®^ Multiplex PCR Premix (Bioneer, Daejeon, South Korea) (Table 3). The amplified products were detected under UV light after the electrophoresis of a 1.5% agarose gel containing 5 μL of RedSafe staining solution (iNtRON, Seongnam, South Korea) and 100 ml of TAE buffer. The product sizes of the specific regions were determined by using a 100 bp DNA ladder (Bioneer) as a size marker.

### 2.4. Specificity and Sensitivity of the Multiplex PCR Assay

Bacteria used to assess the specificity of the multiplex PCR assay were determined using DNA from 27 non-MG strains listed in Table 1. The amplification of the positive control was conducted by mixing DNA templates from each of the four strains (ATCC 15302, ts-11, 6/85, and F strain). The sensitivity of the multiplex PCR assay was evaluated using 5 MG reference strains (ATCC 15302, ts-11, 6/85, F strain, and R-low), 22 MG wild-type strains, and 14 MG field vaccine strains identified by a previously reported GTS assay.

### 2.5. Limit of Detection of the Multiplex PCR Assay

The detection limit of the multiplex PCR assay was determined using 10-fold serial dilutions of a mixture (2.5 ng–2.5 fg/each strain) of 10 ng of the DNA templates from each of four strains (MG ATCC 15302, ts-11, 6/85, and F strain) in a single tube. DNA concentrations were measured using a Nanodrop ND-1000 UV/UVS spectrophotometer (Nanodrop Tech., Wilmington, DE, USA).

### 2.6. Evaluation Using Clinical Samples

The multiplex PCR assay was validated with tracheal tissues and swabs from chickens from four poultry farms. Two broiler breeder farms were unvaccinated, and trachea samples were collected at 61 wks and 66 wks of age. Laying hen and broiler breeder farms were vaccinated with ts-11 and F strain, and samples were collected at 28 wks and 38 wks of age.

## 3. Results

### 3.1. Primer Design

Compared to the 18 strains of MG, a specific region of the ts-11 vaccine strain was confirmed in the interspace section (positions 879439–879607) located from RS03710 to the *rlmB* region. This section showed a similarity of 79–84% with other MG strains, and primer sets were designed for this site. For the 6/85 vaccine strain, a specific site was identified at positions 525779 to 526349 of the *scpA* region. This site showed a similarity of 91% with other MG strains. BLAST analysis of the MGF_RS03965 (positions 598259–598621) region of the F vaccine strain showed that it had 100% similarity with all F strain series (F strain, f99 Avipro, and f99 lab) (Table 2 and Figure 1). In addition, a 245 bp section of the *parE* region of MG species specifically detected MG.

### 3.2. Specificity and Sensitivity of the Multiplex PCR Assay

To determine the specificity of the multiplex PCR assay, we investigated 5 MG strains (ATCC 15302, ts-11, 6/85, F strain, and R-low) and 27 different avian pathogens, as listed in Table 1. As a result, the four primer sets generated specific fragments of 245, 169, 571, and 333 bp for MG species, ts-11, 6/85, and F strain, respectively (Figure 2). 

However, 27 non-MG strains were negative in multiplex PCR assay (Table 1*).* We also used 36 isolates to confirm that the sensitivity of the multiplex PCR method was suitable. MG isolates were classified into 22 field strains and 14 vaccine strains. The fourteen vaccine strains distinctly differentiated one ts-11 strain (no. 40), one 6/85 strain (no. 63), and twelve F strains (nos. 42, 52–55, 57, 62, and 64–68) (Table 1 and Figure 3).

### 3.3. Detection Limit of the Multiplex PCR Assay

The detection limit of the multiplex PCR assay was determined using 10-fold serial dilutions of a mixture of genomic DNA concentrations of the four MG strains (ATCC 15302, ts-11, 6/85, and F strain), ranging from 2.5 ng to 2.5 fg. The assay was able to detect up to 250 fg/μl (Figure 4).

### 3.4. Efficiency on Clinical Samples

To assess the application possibilities of the four primer sets on clinical samples, DNA extracted from the tracheal tissues and swabs from chickens from four poultry farms was applied to the multiplex PCR assay. As a result, vaccine strains were detected in vaccinated farms, and wild-type strains were detected in unvaccinated farms (Figure 5).

## 4. Discussion

MG is considered to be an important bacterial pathogen causing respiratory disease in chickens [4,15]. Once the flocks on farms are infected with this pathogen, they cannot be easily replaced with mycoplasma-free flocks. In particular, MG continues to appear on farms with poor sanitation or dirty environments and on farms of multi-age herds. The use of MG vaccines is perhaps the most effective and efficient means of preventing poultry farms from exposure to the aforementioned risk factors. In particular, the three kinds (ts-11, 6/85, and F strain) of MG vaccines have been widely used in many other countries, including the Netherlands, Asia (South Korea, Thailand, and Jordan), Australia, and Latin America (Mexico and Brazil) [3,14,15,16,17,18].

However, farm managers administer antibiotics to maintain vaccinated flocks or control other respiratory diseases. For this reason, frequent administration of antibiotics may increase resistance rates or interfere with the survival of vaccine strains.

Therefore, early detection of MG-contaminated sources in poultry farms can play an important role in effectively maintaining healthy flocks [18]. The assays applied for monitoring MG currently include bacterial isolation and serological and genetic assays. However, serological methods widely used for diagnosis lack the ability to differentiate between antibodies elicited by natural infection and those elicited by vaccination in vaccinated flocks. In comparison, culture for MG isolation can require at least 21 days and may be inhibited due to the rapid proliferation of other bacteria. Molecular genetic assays (GTS and MLST) could help effectively distinguish the three vaccine strains through sequencing, but doing so takes a long time [13,14,19]. Additionally, conventional PCR and real-time PCR are frequently used instead of culture to detect specific avian mycoplasma DNA, but these two PCR methods can only detect avian mycoplasma species or allow for the simultaneous detection of MG and MS [3,10,20]. Therefore, the new multiplex PCR assay in this study was developed to quickly and simply detect the three MG vaccine strains and wild-type strains simultaneously. Four target genes, the *parE* gene of MG species, the RS03710-*rlmB* interspace section of ts-11, the *scpA* gene of 6/85, and the MGF_RS03965 region of F strain, were specifically identified by the multiplex PCR assay in this study. The detection limit of our novel multiplex PCR assay was very low at 250 fg. Previous reports showed that the detection limit of real-time PCR described by Kahya et al. was 0.9 pg/μL [10]. However, the detection limit of a duplex PCR assay for MG and MS reported by Yadav et al. was 125 ng/mL [20]. Our novel multiplex PCR assay showed a lower detection limit than the real-time PCR and the duplex PCR methods reported previously. This multiplex PCR assay was optimized with a specialized PCR enzyme used in this study. The use of other PCR mixes diminishes sensitivity for a higher detection limit.

In conclusion, the newly developed multiplex PCR technique could be a very useful tool for finding field strains in vaccinated flocks. Moreover, this assay may be helpful as a tool to track specific immunity to *Mycoplasma gallisepticum*. In addition, this multiplex PCR method can replace the existing GTS technique because it can differentiate MG strains within a few hours using four primer sets.

## Figures and Tables

**Figure 1 pathogens-12-00111-f001:**
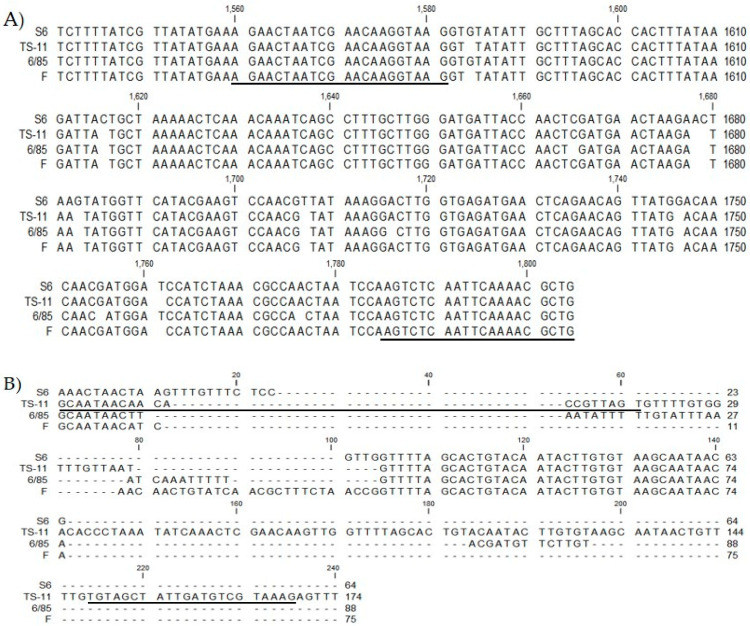
Sequence alignments of four regions specific to each of the four different strains (MG 5302, ts-11, 6/85, and F strain). (**A**) A sequence alignment for MG common gene. (**B**) Specific region of ts-11. (**C**) Specific region of 6/85. (**D**) Specific region of F strain was not present in other MG strains.

**Figure 2 pathogens-12-00111-f002:**
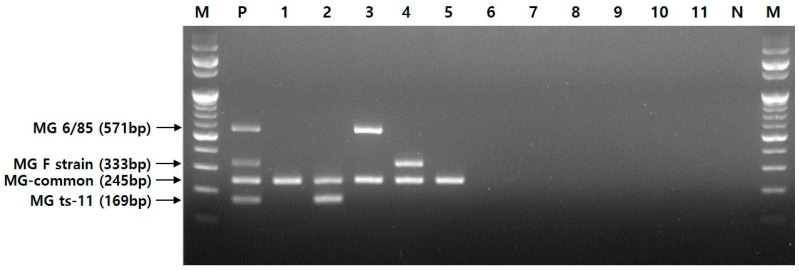
MG species and representative bacteria tested using a novel multiplex PCR assay. Lane M, 100 bp DNA ladder; lane P, positive; lane 1, MG ATCC 15302; lane 2, MG ts-11 (vaccine); lane 3, MG 6/85 (vaccine); lane 4, MG F strain (vaccine); lane 5, MG R-low; lane 6, *M. synoviae*; lane 7, *M. hyopneumoniae*; lane 8, *M. hyorhinis*; lane 9, *A. avium*; lane 10, *E. coli* ATCC 25922; lane 11, *P. canis*; lane N, negative.

**Figure 3 pathogens-12-00111-f003:**
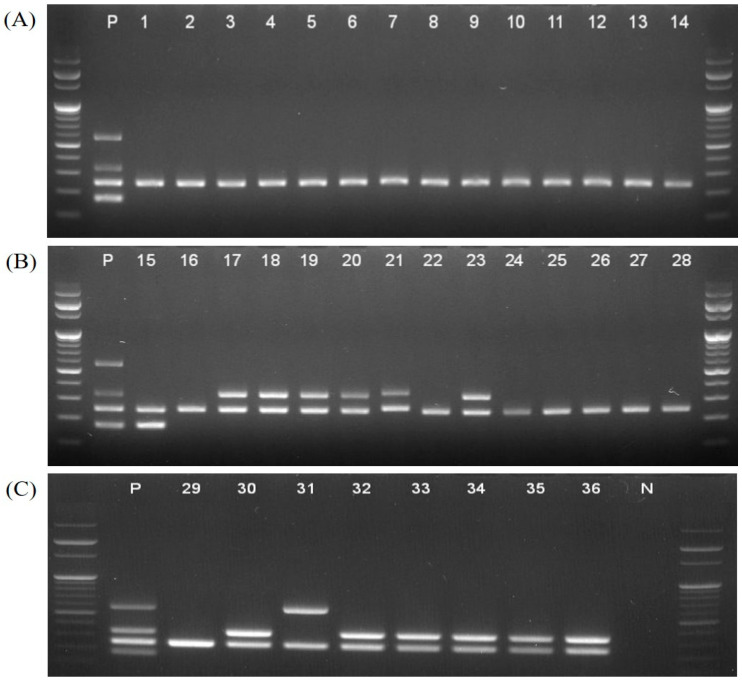
Amplicons of 36 isolates tested using the multiplex PCR assay. (**A**) Wild-type strains (nos. 1–14), (**B**) one ts-11 strain (no.15), six F strains (nos. 17–21, and 23) and seven wild-type strains (nos. 16, 22 and 24–28), and (**C**) one wild-type strain (no. 29), one 6/85 strain (no.31) and six F strains (nos. 30 and 32–36).

**Figure 4 pathogens-12-00111-f004:**
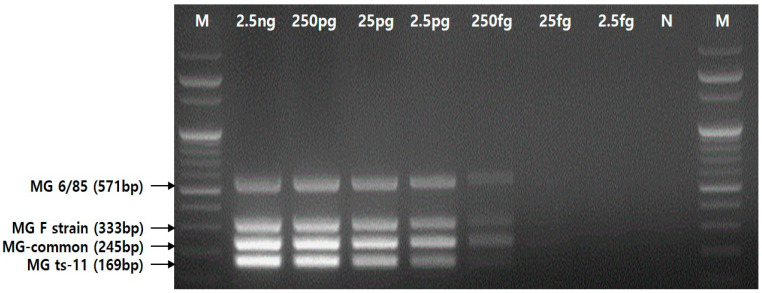
The detection limit of the multiplex PCR assay using the four primer sets.

**Figure 5 pathogens-12-00111-f005:**
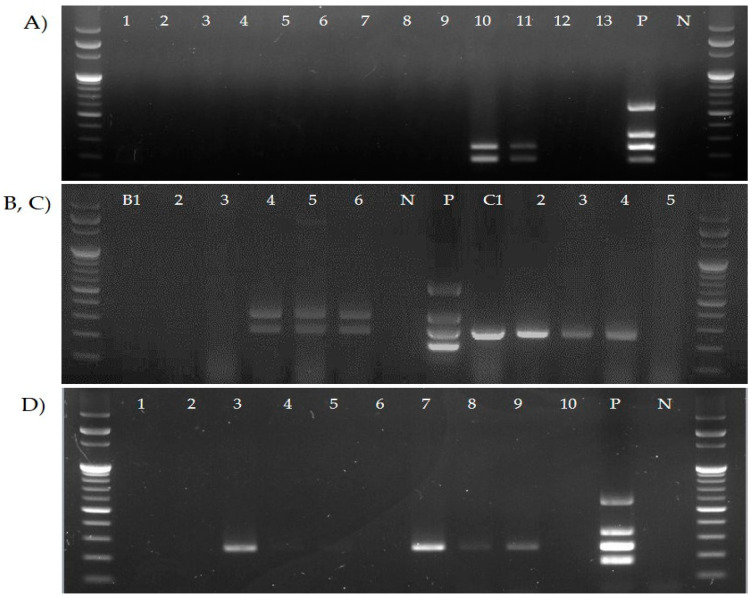
Efficiency of the newly developed multiplex PCR assay on clinical samples. (**A**) Laying hen farm vaccinated with ts-11, (**B**) broiler breeder farm (no. B1–6) vaccinated with F strain, and (**C**,**D**) non-vaccinated broiler breeder farm (no. C1–5 and D1–10).

**Table 1 pathogens-12-00111-t001:** Bacterial strains used to test the sensitivity and specificity of the multiplex PCR assay.

No.	Species	Strains	No.	Species	Strains	No.	Species	Strains
1	*M. gallisepticum*	ATCC15302	24	*S.* Typhimurium	ATCC14028	47	*M. gallisepticum* isolate strain	HS-3
2	*M. gallisepticum*	ts-11	25	*S.* Pullorum	ATCC10398	48	*M. gallisepticum* isolate strain	58–6
3	*M. gallisepticum*	6/85	26	*S.* Gallinarum	ATCC98252	49	*M. gallisepticum* isolate strain	LYK
4	*M. gallisepticum*	F	27	*S. aureus*	ATCC25923	50	*M. gallisepticum* isolate strain	11KCG-51
5	*M. gallisepticum*	R-low	28	*P. aeruginosa*	APQA	51	*M. gallisepticum* isolate strain	11KCH-3
6	*M. synoviae*	NCTC10124	29	*C. coli*	ATCC33559	52	*M. gallisepticum* isolate strain	15JI240
7	*M. hyorhinis*	APQA	30	*C. jejuni*	ATCC33560	53	*M. gallisepticum* isolate strain	16DY-26
8	*M. hyopneumoniae*	ATCC25934	31	*C. perfringens* type A	APQA	54	*M. gallisepticum* isolate strain	16AD084
9	*A. avium*	NCTC11297	32	*C. perfringens* type C	APQA	55	*M. gallisepticum* isolate strain	16AD099–4
10	*A. gallinarum*	NCTC11188	33	*M. gallisepticum* isolate strain	Hanpo	56	*M. gallisepticum* isolate strain	16LO8–6
11	*A. volantium*	NCTC3438	34	*M. gallisepticum* isolate strain	DS7–3	57	*M. gallisepticum* isolate strain	16WS3–3
12	*E. coli*	ATCC25922	35	*M. gallisepticum* isolate strain	41–15	58	*M. gallisepticum* isolate strain	16HBM-4
13	*E. coli*	ATCC35218	36	*M. gallisepticum* isolate strain	6_57	59	*M. gallisepticum* isolate strain	16KWY-27
14	*E. coli*	ATCC8739	37	*M. gallisepticum* isolate strain	11AGH	60	*M. gallisepticum* isolate strain	6KSM-1
15	*E. faecalis*	ATCC29212	38	*M. gallisepticum* isolate strain	NSG	61	*M. gallisepticum* isolate strain	17MHJ-24
16	*E. faecium*	APQA	39	*M. gallisepticum* isolate strain	LJH4	62	*M. gallisepticum* isolate strain	19OS-1
17	*P. multocida* subsp. *gallicida*	ATCC51689	40	*M. gallisepticum* isolate strain	11CSG-17	63	*M. gallisepticum* isolate strain	20HS-G1
18	*P. multocida* subsp. *multocida*	ATCC43137	41	*M. gallisepticum* isolate strain	SJ-1	64	*M. gallisepticum* isolate strain	AD58–1
19	*P. canis*	ATCC43326	42	*M. gallisepticum* isolate strain	13AD117	65	*M. gallisepticum* isolate strain	AD58–2
20	*G. anatis*	NCTC11413	43	*M. gallisepticum* isolate strain	16NU-8	66	*M. gallisepticum* isolate strain	AD58–3
21	*R. anatipestifer*	ATCC11845	44	*M. gallisepticum* isolate strain	11KCH-4	67	*M. gallisepticum* isolate strain	AD56–3
22	*R. anatipestifer*	ATCC51412	45	*M. gallisepticum* isolate strain	16BWB-24	68	*M. gallisepticum* isolate strain	DW1–3
23	*S.* Enteritis	ATCC13076	46	*M. gallisepticum* isolate strain	PSB			

**Table 2 pathogens-12-00111-t002:** Primer sets used for multiplex PCR analysis for the differential identification of MG strains.

Target	Specific Region	Primer Set	Sequence	Product Size (bp)
MG	*parE*	MG-L	AGAACTAATCGAACAAGGTAAG	245
		MG-R	CAGCGTTTTGAATTGAGACT	
ts-11	RS03710-*rlmB* (879439–879607)	ts-11-L	GCAATAACAACACCGTTAGT	169
		ts-11-R	CTTTACGACATCAATAGCTACA	
6/85	*scpA* (525779–526349)	6/85-L	CTTGGTTATAGATAATCGTATTTGG	571
		6/85-R	CCCGACTTTGATAATGAAGAAA	
F	MGF_RS03965 (598289–598621)	F-L	CTTATTAAGTCCAATAGCTCCA	333
		F-R	GCCAAATAATATCGATTGGTTG	

**Table 3 pathogens-12-00111-t003:** The optimal amplification conditions for discriminating between field strains and vaccine strains of MG using multiplex PCR.

Classification	Vol. (μL)	Reaction Condition		Cycles
Nuclease-Free Water	13	Initial denaturation	94 °C, 3 min	1
Primer common-L (5 pmole/μL = 5 μM)	0.75			
Primer common-R (5 pmole/μL = 5 μM)	0.75	Denaturation	94 °C, 30 s	30
Primer ts-11-L (10 pmole/μL = 10 μM)	0.75			
Primer ts-11-R (10 pmole/μL = 10 μM)	0.75	Annealing	52 °C, 30 s	
Primer 6/85-L (5 pmole/μL = 5 μM)	0.75			
Primer 6/85-R (5 pmole/μL = 5 μM)	0.75	Extension	72 °C, 1 min	
Primer F-L (5 pmole/μL = 5 μM)	0.75			
Primer F-R (5 pmole/μL = 5 μM)	0.75	Final extension	72 °C, 5 min	1
DNA	1			
Total	20	Cooling	4 °C	∞

## Data Availability

Not applicable.

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
