# Peer review of "A Multiplex PCR Assay for Differential Identification of Wild-type and Vaccine Strains of Mycoplasma gallisepticum"

_pathogens, 2023, doi:10.3390/pathogens12010111_

Round 1

Reviewer 1 Report

Review on

A multiplex PCR assay for differential identification of wild-type and vaccine strain of mycoplasma gallisepticum

This manuscript describes the multiplex PCR determination of Mycoplasma gallisepticum to distinguish the vaccine and pathogen strain, Korean isolates by using 4 genes. This method provide higher sensitivity than previous methods.

1.       To affirm the precision of PCR amplification. The authenticity of each PCR product should be validated by DNA sequencing.

2.       Table 3, the amount of DNA used in optimized PCR reaction should be present in ‘ug or ng’ unit.

3.       In the discussion, the using of this multiplex PCR determination would be provide to other application such as

a.       Following the status of lived vaccine in vaccinated animals especially as a tool to follow the specific immunity to Mycoplasma gallisepticum.

b.       Distinguishing the vaccinated and infected animals.

Author Response

Thank you for your helpful information on the revision of our manuscript.

We revised our manuscript according to the Reviewer’s comments and revised sentences were remarked with red color.

Reviewer 2 Report

The presented manuscript is correctly written. The authors described a method of detection of Mycoplasma gallisepticum (MG), which can be used as a quick method of its detection using multiplex PCR.

The authors used both vaccine species and isolated them from the environment. Only from vaccine species DNA was detected with a detection limit ranging from 2,5 ng to 2.5 fg. No DNA products were detected in non-MG strains. The results of this study confirmed that conventional PCR could be used to identify MG similar to qPCR allowing the detection of the DNA in a very small concentration. 

In my opinion, the presented study does not present originality and novelty. 

Multiplex PCR is routinely used in the microbiological identification of different patogens.

Author Response

(The authors gave the same response as above.)

Reviewer 3 Report

Kang et al, in this paper presents a simple novel PCR based assay to discriminate Wild-type and the commonly used ts-11, 6/85, and F strain vaccine strains of Mycoplasma gallisepticum. The multiplex PCR could detect up 250 fg/μl of MG DNA. Using these MG multiplex primers, no non-specific amplification was detected for other common avian bacterial pathogens. I find the manuscript well written, and the topic is relevant considering the fact that Mycoplasma gallisepticum is a major  pathogen causing significant damages in the poultry industry around the world. This manuscript is recommended for publication after addressing a few critical points listed below to improve it.

Comment 1: Nucleotide alignment of the four gene sequences, to show how the sequences are discriminated by the different primers: It would be appreciated if the authors could provide four different alignment figures displaying the target amplicon sequences for the genes parE (2345 base pairs), RS03710-rlmB (169 base pairs), ScpA (571 base pairs), MGF_RS03965 (333 base pairs), in which the sequences of each of the four different strains MG, TS-11, 6/85 and F strains are aligned. The primer binding site could be marked in the alignment by underline. This alignment figure will serve as a visual guide for a better appreciation of the strain-specific primer design discussed in Results section 3.1

Additionally, it would be helpful if the authors could elaborate on how the primer specificity of amplification is achieved for each vaccine strain/gene. Is this due to a specific stretch of genes/sequences present in the strains or by sequence-specific primer discrimination (primers cannot bind due to sequence mismatch) in the PCR procedure?

Comment 2: Multiplex PCR testing on actual field samples missing: In most of the methods/ results presented, the authors seem to have tested the primers on purified strains or isolates. But in real-world testing, other factors have to be considered – like low abundance of the target, presence of huge amounts of host DNA, normal microflora in the host tissues than can cause mispriming in PCR, presence of PCR inhibitors etc. The authors are suggested to test the primers on actual field samples and compare the results, with regard to sensitivity.  In the manuscript, the actual testing of field samples is one of the key areas that need to be addressed from a nucleic acid-based molecular diagnostics developmental perspective.

In figure 1, the authors seem to have tested the multiplex primers on purified MG DNA from pure isolates of the MG vaccine strains/ wild type MG ATCC 1530. A multiplex PCR analysis of an actual sample (tracheal /cloacal/choanal swabs or other tissues of poultry tested for MG) would be interesting since it would contain the host genetic material as well as other microorganisms that live naturally in the poultry birds. To validate this test in real-life scenarios, it would be helpful if the authors provided a gel image of these multiplex primers tested on actual cloacal/choanal swabs/ other poultry tissues or even spiked samples.

Tests can be conducted on actual field samples to diagnose nonspecific amplification if any, given the 8 PCR primers used in the multiplex and the relatively low annealing temperature of 52°C. If the authors could test this multiplex PCR on actual field samples of known MG+ve birds or swabs from birds vaccinated with TS-11, 6/85 and F strains and provide the picture as an extra panel Figure 1B, that would be great. An MG-ve bird swab/tissue can be kept as a negative control.

Comment 3 : If the authors can include the amplification pictures using the multiplex primers for the various M. gallisepticum isolate strain listed in table 3 (sl numbers 33 to 68), it would improve the manuscript very much.

Comment 4 : Infection of individual birds with more than one Mycoplasma gallisepticum strain has been reported. Can the multiplex PCR detect and differentiate such infections ?. In Figure 1 the authors have tested the multiplex PCR with individual samples. It would be interesting to see if the authors can detect different combinations of more than one strain with multiplex PCR. A small aliquot of the different combinations of individual DNAs (a few important dual strain combinations that could be possibly encountered in a poultry scenario) can be mixed in-vitro and tested by the multiplex PCR to see how these primers detect mixed infections.

Comment 5: The authors have used a specialised PCR enzyme AccuPower Multiplex PCR Premix to optimize the MG multiplex PCR. Would the use of normal PCR mix (Regular Taq DNA polymerase) affect the results, with regard to sensitivity, specificity and non-specific amplification. If the authors could comment on this in the discussion, it would be beneficial to readers.

Author Response

(The authors gave the same response as above.)

Round 2

Reviewer 2 Report

After the authors' proofreading, I accept this manuscript.

Reviewer 3 Report

All queries or comments made during the first round of review have been addressed systematically by the authors. Acceptance of the manuscript for publication is recommended.